# NGS Sequencing Reveals New *UCP1* Gene Variants Potentially Associated with MetS and/or T2DM Risk in the Polish Population—A Preliminary Study

**DOI:** 10.3390/genes14040789

**Published:** 2023-03-24

**Authors:** Anna Andrzejczak, Agata Witkowicz, Dorota Kujawa, Damian Skrypnik, Monika Szulińska, Paweł Bogdański, Łukasz Łaczmański, Lidia Karabon

**Affiliations:** 1Laboratory of Genetics and Epigenetics of Human Diseases, Department of Experimental Therapy, Hirszfeld Institute of Immunology and Experimental Therapy, Polish Academy of Sciences, 53-114 Wroclaw, Poland; 2Laboratory of Genomics and Bioinformatics, Department of Immunology of Infectious Diseases, Hirszfeld Institute of Immunology and Experimental Therapy, Polish Academy of Sciences, 53-114 Wroclaw, Poland; 3Department of Treatment of Obesity, Metabolic Disorders and Clinical Dietetics, Poznan University of Medical Sciences, 60-569 Poznan, Poland

**Keywords:** UCP1, thermogenin 1, NGS, polymorphism, SNV, MetS, T2DM, disease risk

## Abstract

The number of people suffering from metabolic syndrome (MetS) including type 2 diabetes (T2DM), hypertension, and obesity increased over 10 times through the last 30 years and it is a severe public health concern worldwide. Uncoupling protein 1 (UCP1) is a mitochondrial carrier protein found only in brown adipose tissue involved in thermogenesis and energy expenditure. Several studies showed an association between *UCP1* variants and the susceptibility to MetS, T2DM, and/or obesity in various populations; all these studies were, however, limited to a few selected polymorphisms. The present study aimed to search within the entire *UCP1* gene for new variants potentially associated with MetS and/or T2DM risk. We performed NGS sequencing of the entire *UCP1* gene in 59 MetS patients including 29 T2DM patients, and 36 controls using the MiSeq platform. An analysis of allele and genotype distribution revealed nine variations which seem to be interesting in the context of MetS and fifteen in the context of T2DM. Altogether, we identified 12 new variants, among which only rs3811787 was investigated previously by others. Thereby, NGS sequencing revealed new intriguing *UCP1* gene variants potentially associated with MetS and/or T2DM risk in the Polish population.

## 1. Introduction

The term metabolic syndrome (MetS) is defined as the co-occurrence of related metabolic disorders such as elevated blood glucose levels, hypertension, lipid disorders, elevated body mass index (BMI), abdominal obesity, and insulin resistance, among others. These disorders promote the development of cardiovascular disease (CDV) (stroke, coronary heart disease, and myocardial infarction) and type 2 diabetes (T2DM) [1,2].

The WOBASZ II study, published in 2021, indicated that in Poland, the prevalence of MetS was 32.8% in women and 39% in men [3,4]. Moreover, when compared to the previous study (WOBASZ I), a significant increase in the prevalence of MetS in Polish adults (aged 20–74) was observed (3.3% in women and 8.8% in men) in only one decade. This was due to the increased frequency of elevating fasting glucose or diabetes, abdominal obesity (especially in women), and lipid disorders (mainly in men) [3,4].

Despite advances in diagnostics, treatment, and prevention, the prevalence of CVD reportedly increases year after year worldwide [5], and this disease remains the leading cause of death in developed countries. An estimated 17.9 million people died from CVD in 2019, representing 32% of all global deaths [6]. The prevalence of diabetes also increased markedly worldwide [5]. In Poland, the number of people with T2DM grows every year. In 2014, the estimated incidence of diabetes was 2.113 million, increasing to 2.533 million in 2017 [7]. The increase in T2DM was positively correlated with the growing number of overweight and obese individuals.

Abdominal obesity and the associated excess of visceral fat is a cause of disorders included in MetS. It is well known that obesity is caused by prolonged unbalanced caloric intake in relation to energy expenditure. Total body energy expenditure represents the conversion of oxygen and food (or storable forms of energy—triacylglycerols accumulated in adipose tissue), to carbon dioxide, water, heat, and “work” into the environment. Energy expenditure in humans can be subdivided into: (1) basal energy expenditure required for normal cell functioning; (2) energy expenditure resulting from physical activity; and (3) energy expenditure attributed to adaptive thermogenesis [8].

Two kinds of adipose tissues are presented in the human body: white and brown adipose tissue [9]. The most important function of white adipose tissue is energy storage in the form of triacylglycerols, while brown adipose tissue (BAT) is responsible for the thermogenesis process combined with heat release due to the dissipation of chemical energy [8]. The size and thermogenic activity of BAT depend on the age, sex, season, body mass index (BMI), and the level of physical activity [10]. Several studies showed that, even though the contribution of BAT to total energy expenditure is only 1–5% of the resting metabolic rate (RMR), with appropriate stimulation, this value can increase to 16% of the RMR [11]. BAT activity is lower in obese people than in lean people, as BAT is inversely proportional to BMI [12,13]. Therefore, interest in this tissue increased significantly in the last decade due to its potential role in preventing obesity and obesity-related diseases.

The key protein for BAT functioning is the uncoupling protein 1 (UCP1) called thermogenin 1, located in the inner mitochondrial membrane [14]. This protein is responsible for the conversion of excess fat into heat energy. UCP1 transports H^+^ across the inner mitochondrial membrane in presence of long-chain fatty acids, which makes BAT mitochondria produce heat at the expense of ATP [15]. *UCP1* gene expression is increased by cold, adrenergic stimulation, β3-agonists, retinoid and thyroid hormones, and cAMP, while its expression is activated by non-esterified fatty acids and inhibited by purine nucleotides (GDP, ATP, and ADP) [16]. Taking into consideration the role of UCP1 in BAT thermogenesis as well as in increasing energy expenditure, it is postulated that its activity might be related to obesity.

Numerous studies showed that T2DM and obesity are multifactorial disorders where both genetic and environmental backgrounds play a crucial role in susceptibility determination [17]. Great efforts were made to identify the genes associated with these conditions, with particular focus given to the genes related to energy expenditure, including genes encoding adrenergic receptors and mitochondrial uncoupling proteins (UCPs) [18]. Taking into consideration the role of UCP1 in thermogenesis in BAT as well as in increasing energy expenditure, it is postulated that polymorphisms existing within the *UCP1* gene might be associated with deficiency of UCP1 protein levels and activity, and may be a risk factor for MetS and related disorders, especially T2DM. Therefore, numerous genetic studies investigated the association between *UCP1* gene single nucleotide variations (SNVs) and the risk of obesity, T2DM, and MetS. However, only a limited number of SNVs selected mainly on the basis of previous literature data were investigated. The most common polymorphisms studied in relation to MetS, T2DM, obesity, BMI, and lipid disorders were: –3826A/G (rs1800592), –112A/C (rs10011540), and –1766A/G (rs3811791) in the 5′-region, Ala64Thr (rs45539933) in exon 2, and Met229Leu (rs2270565) in exon 5 of the *UCP1* gene. These were described in great detail in reviews by Brondani et al., Stosio et al, Dinas et al., and Fluoris et al. [19,20,21,22,23].

Numerous literature data showed the associations of *UCP1* variants with MetS; however, only a limited number of arbitrarily selected polymorphisms were investigated. These facts prompted us to search deeper for variability within the coding and regulatory regions of the *UCP1* gene. Therefore, the aim of the presented study was to search, using next generation sequencing (NGS) technology, for new, not previously studied *UCP1* gene variants, that are potentially associated with MetS and/or T2DM risk.

## 2. Materials and Methods

### 2.1. Patients

For the purpose of this study, a group of 59 patients was recruited based on the following criteria: (1) of adult age; (2) diagnosed with MetS according to IDF (International Diabetes Federation) guidelines; (3) having central obesity, defined as waist circumference ≥ 80 cm in women or ≥ 94 cm in men; and (4) having at least two of the following factors: (i) triglycerides level ≥ 150 mg/dL (1.7 mmol/L) or treatment specific for this abnormality; (ii) HDL level < 40 mg/dL (1.03 mmol/L) in men or < 50 mg/dL (1.29 mmol/L) in women or treatment specific for this abnormality; (iii) systolic blood pressure ≥ 130 mmHg or diastolic blood pressure ≥ 85 mmHg, or treatment of diagnosed hypertension; (iv) fasting blood glucose ≥ 100 mg/dL (5.6 mmol/L) or previously diagnosed T2DM. Among the selected patients, 29 of them also suffered from T2DM. Patient samples were collected during the project U-GENE “Multi-national network of excellence for research on genetic predisposition to cardio-metabolic disorders due to *UCP1* gene polymorphisms” (European Union 7th Framework Program—FP7-PEOPLE-2013-IRSES Grant No. 319010) from 2013 to 2017. The study was conducted in accordance with the Declaration of Helsinki and was approved by the Institutional Review Board Bioethics Committee at Poznan University of Medical Sciences (KB 215/13, annexed 85/16). The patients provided their written informed consent to participate in this study. Patient characteristics are presented in Table 1.

### 2.2. Controls

The control group consisted of 36 participants who were derived from the same geographical area as the patients. Healthy participants were asked to fill in questionnaires where they provided information about age, sex, smoking status, and any present illnesses. Additionally, on the day of blood collection, parameters such as height, weight, and waist circumference were measured, and WHR and BMI were calculated. The control group were selected based on the following criteria: (1) of adult age; (2) BMI ≤ 25; (3) non-smoking; (4) body fat percentage below 35% for women, and below 30% for men; and (5) no diagnosis of hypertension, elevated fasting glucose or diabetes, or thyroid diseases (Table 1). The participants provided their written informed consent to participate in this study.

### 2.3. DNA Isolation and Genotyping

Genomic DNA was isolated from refrozen blood samples by Invisorb QIAamp DNA Blood Mini Kit (Qiagen, Hilden, Germany) or Spin Blood Mini Kit (Stratec Molecular, Birkenfeld, Germany) according to the manufacturer’s instructions. The quality and quantity of the DNA samples were determined using DS-11 spectrophotometer (DeNovix, Wilmington, DE, USA) and Qubit 3.0 Fluorometer (ThermoFisher Scientific, Waltham, MA, USA) using Qubit dsDNA HS Assay (ThermoFisher Scientific, Waltham, MA, USA). Integrity of DNA was determined by capillary electrophoresis on TapeStation 4200 (Agilent Technologies, Santa Clara, CA, USA) using Genomic DNA ScreenTape Assay (Agilent Technologies, Santa Clara, CA, USA).

### 2.4. Library Preparation and Next Generation Sequencing

High quality DNA was selected for NGS sequencing. We performed targeted panel sequencing to identify genetic variants within the *UCP1* gene for MetS patients and controls. To build amplicon sequencing libraries for the coding region of the *UCP1* gene, we used TruSeq Custom Amplicon Low Input Kit (Illumina, San Diego, CA, USA) with TruSeq Dual Index Sequencing Primers (Illumina, San Diego, CA, USA). For 3′UTR and 5′UTR regions of the *UCP1* gene, we used AmpliSeq Illumina Custom DNA Panel (Illumina, San Diego, CA, USA) with AmpliSeq CD Indexes (Illumina, San Diego, CA, USA). The quality and quantity of the purified libraries was assessed by Qubit 3.0 Fluorometer (ThermoFisher Scientific, Waltham, MA, USA) using Qubit dsDNA HS Assay (ThermoFisher Scientific, Waltham, MA, USA) and TapeStation 4200 (Agilent Technologies, Santa Clara, CA, USA) using High Sensitivity D1000 ScreenTape Assay (Agilent Technologies, Santa Clara, CA, USA). Only high-quality libraries were sequenced on the MiSeq System (Illumina, San Diego, CA, USA) using the V2 Illumina Sequencing Kit 500 (Illumina, San Diego, CA, USA).

### 2.5. NGS Data Analysis

FASTQ-formatted sequences were analyzed for quality control by FASTQC open-source software [24]. Reads with quality scores below 24 were excluded. The minimum coverage was 200. Bowtie2 software (ver. 2.5.0) [25] was used to align reads to reference the human genome (Hg38). BAM files were visualized by IGV software (ver. 2.14) [26]. Variant annotation was undertaken using SnpEff (ver. 5.1) [27] and ANNOVAR software (ver. 2019Oct24) [28].

To validate the obtained NGS sequencing results, we compared the results of four *UCP1* polymorphisms previously genotyped (using TaqMan probes) by us: rs1800592, rs10011540, rs45539933, and rs2270565. The degree of concordance between the results was 100%.

To identify the rsIDs and frequencies of considered *UCP1* gene variants, we used USCS and dbSNV databases (available at https://genome.ucsc.edu/index.html and https://www.ncbi.nlm.nih.gov/SNV/, accessed on 24 May 2021). In short, the chromosome position of the variant was pasted to the USCS browser with the open track for Short Genetic Variants from dbSNV release 155 and 153. Next, if the rsID for the searched position was present, the rsID was pasted into the dbSNV browser to retrieve more information about the specific SNV including variation frequency, localization within the gene, variation type, and variation consequence. Additionally, selected SNVs were searched for in the VarSome database (available at https://varsome.com/, accessed on 24 May 2021) for ACMG Classification and Conservation Score (phyloP100).

The difference between allele and genotype distribution in the patient and control groups was analyzed with χ2 tests. Odds ratios (OR) and 95% confidence intervals (95% CI) were calculated using binary logistics regression model to evaluate the relationship between the studied gene variants and susceptibility to MetS. Differences between groups were considered statistically significant if *p* < 0.05. Statistical analysis was conducted using Statistica 13.1 (TIBCO, Inc., Palo Alto, CA, USA).

Linkage disequilibrium (LD) analysis was prepared using the online LDpair tool (available at https://ldlink.nci.nih.gov/?tab=ldpair, accessed on 24 May 2021), which allows for the investigation of correlated alleles for a pair of variants in high LD based on data available on dbSNV. LDpair analysis was performed on GRCh38 High Coverage (1000 Genomes Project) in the CEU (Utah residents with Northern and Western European ancestry) population. The correlation of alleles for two genetic variants was calculated by R^2^ method for *UCP1* gene variants with rsIDs. Gene variants with a frequency below 1% in the European population were excluded from LD analysis due to their rarity. Additionally, we calculated LD for SNVs associated with MetS and/ or MetS with T2DM using SHEsis software [29] on the data from the control group.

## 3. Results

### 3.1. UCP1 Gene Variants Identified by NGS Sequencing

Using NGS sequencing, we identified 85 sequence variations of the *UCP1* gene in controls and MetS patients in total (Appendix A). In detail, thirty-five gene variants were located in the 5′UTR region, forty-three in the intronic regions, six in the exonic regions, and three in 3′UTR region. Based on data from USCS and dbSNV, we were able to identify 59 variants from the detected 85 sequence variations. Between the detected variants, SNVs constituted the largest group. Among them, based on dbSNV, 14 were found to have a frequency of less than 1% in the European population.

Analysis of allele and genotype distribution revealed nine variations which seem to be interesting in the context of MetS risk. For 7 of those variants, we were able to identify the rsIDs. Their locations were as follows: one within 3′UTR region (rs77149926), three within coding region (chr4:140560701, rs183105785, chr4:140563477), and five within 5′UTR region (rs36207410, rs36207408, rs76129861, rs56724370, rs77178927) of the *UCP1* gene. For the following SNVs: rs77149926, rs36207410, rs36207408, rs76129861, rs56724370, and rs77178927 minor alleles were significantly less frequent in MetS patients than in the control group, while for those: chr4:140560701, rs183105785, and chr4:140563477 minor alleles were significantly more frequent in MetS patients compared to the control group (Table 2 and Appendix A).

Furthermore, we divided the MetS group into MetS without T2DM and MetS with T2DM to see if the presence of a specific UCP1 gene variant can be associated with T2DM. In MetS patients with T2DM, we observed 15 interesting UCP1 gene variants in the context of T2DM susceptibility (Table 3). Out of 15 variants, we identified rsIDs for eight positions. Among them, one appeared to be an insertion (rs3138733).

The location of them were as follows: one SNV was located within 3′UTR region (rs77149926), nine within coding region (chr4:140560701, chr4:140561203, chr4:140561340, rs183105785, chr4:140563477, chr4:140564690, chr4:140567294, rs3138733, and chr4:140568337), and five within 5′UTR region (rs3811787, rs36207410, rs6536992, rs6536993, and rs6536994) of the *UCP1* gene.

The following variations were significantly less frequent in MetS with T2DM patients than in the control group: rs77149926, rs3138733, and rs36207410, whereas variations: chr4:140560701, chr4:140561203, chr4:140561340, rs183105785, chr4:140563477, chr4:140564690, chr4:140567294, chr4:140568337, rs3811787, rs6536992, rs6536993, and rs6536994 were significantly more frequent in T2DM patients compared to controls (Table 3).

The distribution of genotypes and alleles in MetS patients without T2DM did not differ significantly from that observed in controls (Appendix A).

As presented in Table 4, five variants were differently distributed in the whole group of MetS patients as well as MetS with T2DM compared to controls. Furthermore, we observed variants which were distributed differently as compared to controls only in MetS patients with T2DM, especially within the coding and 5′UTR region of the *UCP1* gene.

Altogether 12 variants, for which distribution of allele and genotypes in MeS and/or MetS with T2DM and controls were different, were found to have rsIDs. To see which variants were previously studied by others, we searched every variant in dbSNV and VarSome databases and investigated the publications tab. In this analysis, we focused on the variants found in our study: rs77149926, rs183105785, rs3138733, rs3811787, rs36207410, rs6536992, rs6536993, rs6536994, rs36207408, rs76129861, rs56724370, and rs77178927. Our analysis of dbSNV revealed that only one of these variants was studied previously by others (rs3811787) (Table 4). In the case of the remaining 11 variants, there were no articles studying their correlation with MetS and/or T2DM risk (Table 4).

In summary, five SNVs were associated with the susceptibility to MetS in the whole group of patients as well as MetS with T2DM, ten were associated only with MetS with T2DM, while four SNVs were significantly associated with MetS in the whole group of patients. Among them, only rs3811787 was investigated previously. However, we would like to draw attention to the fact that the results achieved should be treated with caution, since the compared groups were small and the results of these analyses can only be treated as direction for future investigations.

### 3.2. LD Analysis of SNVs Idetyfied by NGS Sequencing

Taking into consideration the limitation of the LDpair tool, analysis was performed only for the identified *UCP1* gene variants (established rsIDs) with a minor allele frequency (MAF) higher than 1% in the European population, and also, the variant must be present on the LDpair tool. We performed LD analysis separately for variants located within regulatory regions (3′UTR and 5′UTR) and coding regions of the *UCP1* gene.

In the case of coding regions based on the results from the LDpair software, we identified two very strong LD blocks with r^2^ = 1. The first LD block consisted of four variants: rs726989, rs2043124, rs11932232, and rs6818140, and the second LD block comprised two variants: rs2270565 and rs45539933 (Figure 1).

The LD analysis of regulatory regions showed that there were four very strong LD blocks with r^2^ = 1 in the 5′UTR region of the *UCP1* gene. The first LD block consisted of three variants: rs10011540, rs3749539, and rs3811789; the second LD block comprised six variants: rs36207410, rs36207408, rs76129861, rs56724370, rs77178927, and rs79430751; the third LD block included five variants: rs1430579, rs1430578, rs1472268, rs1472269, and rs1800592; and the last LD block contained three variants: rs6536992, rs6536993, and rs1434525671. In the 3′UTR region, there were no significant LD blocks (Figure 2).

The analysis performed with the use of SHEsis software on the genotyping data for SNVs potentially associated with MetS or MetS with T2DM (Table 4) showed that all SNVs, except one polymorphism (rs3811787) located in the regulatory region of UCP1 gene, were in the strong LD (r^2^ = 1). However, in the coding region, only one LD block including chr4:140561340, chr4:140564690, and chr4:140567294 with moderate LD (r^2^ ≈ 0.7) was observed (Appendix A).

## 4. Discussion

For years, the treatment of chronic metabolic diseases and obesity focused on developing a properly balanced diet and increasing energy consumption through increased physical exercise. In recent years, multiple studies documented that MetS, including obesity and T2DM, are multifactorial diseases whose development is not solely determined by environmental factors. Numerous studies showed that multiple biochemical and genetic factors play an important role in MetS pathogenesis [33,34]. Among others, it was demonstrated that all three members of the peroxisome proliferator-activated receptor (PPAR) nuclear receptor subfamily, PPARα, PPARβ/δ, and PPARγ, are critical in regulating insulin sensitivity, adipogenesis, lipid metabolism, and blood pressure. Recently, several studies indicated that SNVs of PPARs, such as Leu(162)Val and Val(227)Ala of PPARα, +294T > C of PPARβ/δ, Pro(12)Ala, and C1431T of PPARγ, are significantly associated with the onset and progression of MetS and T2DM in different populations worldwide. Furthermore, it was demonstrated that the glucose metabolism and lipid metabolism were influenced by gene–gene interaction among PPARs genes [35,36].

On the other hand, it is postulated that abnormal metabolism of BAT and disturbed thermogenesis led to obesity in mice [37,38]. This observation attracted the interest of scientists to the study of the potential role of BAT in the development of MetS. Attention was drawn to genes encoding the UCP1 protein and adrenergic receptors as genes potentially being associated with the predisposition to MetS and related diseases.

The transcriptional regulation of the gene is located in the 5′ non-coding region. In the *UCP1* gene, this region includes a proximal regulatory region, immediately upstream of the transcription start site that contains a CREB (cAMP response element-binding protein) binding site, which mediates a positive transcriptional response to cAMP and a negative response to AP2 (c-Jun/c-Fos) complexes, and a C/EBP (CCAAT-enhancer-binding protein) binding site capable of responding to C/EBPa and C/EBPb. Recently, the transcription factor zinc finger protein-516 (Zfp516) was shown to bind to the proximal region of the *UCP1* gene promoter to play a role in cold-induced *UCP1* gene expression. In addition to the proximal region, there is a strong enhancer region that contains a cluster of response elements for nuclear hormone receptors that bind PPARα/retinoid X receptor (RXR), PPARγ/RXR, RAR/RXR, and thyroid receptor (TR)/RXR heterodimers, providing responsiveness to ligands of PPARγ and PPARα (e.g., naturally occurring fatty acid derivatives, specific drugs such as fibrates and thiazolidinediones), retinoic acids, and thyroid hormones. Moreover, in the *UCP1* promoter, the activating transcription factor-2 (ATF2)-binding site that provides cAMP-dependent responsiveness to CP1 gene transcription is also present. Transcriptional coregulators, such as the PPAR-g coactivator 1-a (PGC-1a), bind and regulate PPAR/RXR heterodimers and possibly other nuclear receptor dimer complexes. It postulated that the enhancer region is strongly involved in determining BAT-specific transcription of the *UCP1* gene [16].

To date, only a limited number of genetic variants of the *UCP1* gene were investigated previously, selected mainly on the basis of literature data. The most commonly studied SNVs were: rs1800592 (–3826A/G), rs10011540 (–112A/C), and rs3811791 (–1766A/G) in 5′UTR region, rs45539933 (Ala64Thr) in exon 2, and rs2270565 (Met229Leu) in exon 5 of the *UCP1* gene. Numerous studies concerned the association of *UCP1* polymorphisms with susceptibility to MetS, obesity, and T2DM [21,22,23]. In those studies, the relationship of –3826A/G, –1766A/G, Met229Leu, and Ala64Thr with abnormalities in *UCP1* mRNA expression [39,40] increased BMI [41,42], unfavorable LDL/HDL ratio [18], and increased glucose [43] was shown.

Given the above, in the present study, we sequenced the entire *UCP1* gene using NGS technology to search for new and potential disease risk markers that may be worth studying in depth in the future. We found that five *UCP1* variants were differently distributed in the overall group of MetS patients, as well as MetS with T2DM compared to controls. Among them, the chr4:140563477 variation is located in exon 3 of the *UCP1* gene. This variation leads to the exchange of the allele T to C resulting in a missense mutation, causing substitution from threonine (T) to alanine (A) in the position 123 (T123A). Threonine is a polar amino acid, whereas alanine is a non-polar hydrophobic amino acid. Therefore, this T123A modification may have an influence on proper UCP1 protein structure.

Furthermore, we observed SNVs which were distributed differently compared to controls only in the MetS patient group with T2DM. rs6536992, rs6536993, and rs6536994 are located in the 5′UTR region and situated in potential binding sites for transcription factors such as Prdm5, ERS1, RUNX2, PRDM9, MEF2C, and TEAD4 (based on JASPAR Transcription Factor Binding Site Database in UCSC). Therefore, the presence of those variants within the 5′ enhancer or promotor regions may disturb the binding of specific transcription factors.

Another SNV of interest in this study, rs3811787 located in the 5′UTR region, was the only SNV indicated here that was previously described in the literature. The association between rs3811787 and diabetic retinopathy risk in T2DM was investigated by Jin et al. [31]. It was shown that in patients with uncontrolled blood glucose, the rs3811787 T allele was associated with a decreased risk of diabetic retinopathy. In a subsequent study in Korean women, Cha et al. found an association between rs3811787 and rs1800592 and body fat accumulation and obesity [30]. Moreover, Nikaranova’s study [32] showed the influence of rs3811787 on the leptin-mediated thermoregulation mechanism. In this study, the prevalence gradient of the rs3811787T allele of *UCP1* increased from the south to the north across Eurasia, along the shore of the Arctic Ocean. Thereby, the authors suggest the potential involvement of the *UCP1* gene, in particular the rs3811787 T allele, in better activation of the *UCP1* gene and higher activity of BAT [32]. Our very preliminary data also confirmed the decreased risk of T2DM individuals possessing the rs3811787 TT genotype.

In summary, our study revealed new and interesting genetic variations of the *UCP1* gene worthy of studying in the future in the context of MetS and/or T2DM susceptibility. Most of the variants identified were located in the regulatory regions of the *UCP1* gene suggesting their potential role in the regulation of UCP1 expression. However, further studies are needed to elucidate their role in MetS and T2DM susceptibility.

The main limitation of this study was the small sample size of the sequenced MetS and control group, which made it impossible to draw far-reaching conclusions in regard to disease risk. However, the aim of this study was only to search for new or interesting disease markers in the context of MetS and/or T2DM disease. The results obtained serve as preliminary research for wider projects, in which we plan to investigate the association of the new *UCP1* gene variants discovered here that may be potentially associated with MetS and/or T2DM susceptibility in much bigger groups of MetS and T2DM patients. Secondly, the library of coding and regulatory regions (5′UTR and 3′UTR) were prepared using two different Illumina kits (TruSeq and AmpliSeq). The first algorithm diagnosed coding regions (TrueSeq) based on specific probes, while the second (AmpliSeq) was based on the sequencing of amplicons. Therefore, theoretically, the results may differ in the frequency of identified variants. However, the library for the fragment of the 5′ UTR region was prepared using both TruSeq and AmpliSeq kits, and NGS results were in full compliance. The third limitation is the mismatch of ages between the older MetS and the younger control groups. Fortunately, the potential of the control subjects to develop MetS later in their lives should lead to underestimation of their enhancing or protective effects, which, therefore, should leave our conclusions largely intact. 

## 5. Conclusions

In conclusion, to date, only a limited number of genetic variants present within the *UCP1* gene were studied (rs1800592, rs10011540, rs3811791, rs45539933, and rs2270565). Our preliminary NGS data showed that many other genetic variants were present within the *UCP1* gene. Among the identified SNVs, some of them are worth taking into consideration when studying the susceptibility to MetS and/or T2DM. Moreover, we identified a few LD blocks, which enables a more economical approach to further research by selecting the tag SNVs representing each LD block. Therefore, the presented results might be treated as directions for future studies on the association between *UCP1* gene SNVs and MetS/T2DM risk on larger groups of cases and controls in different populations.

## Figures and Tables

**Figure 1 genes-14-00789-f001:**
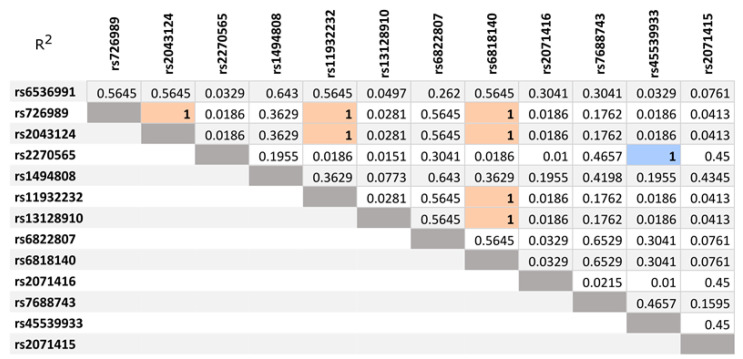
LD analysis of *UCP1* gene variants identified in the coding region based on data from the LDpair tool. LD blocks are marked by colors.

**Figure 2 genes-14-00789-f002:**
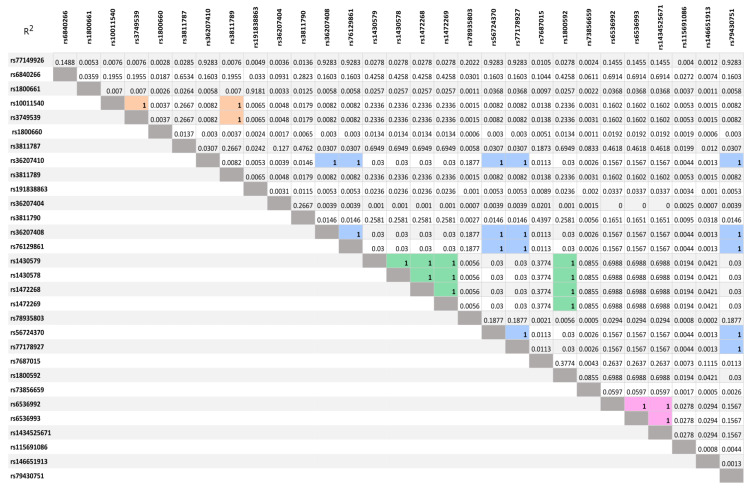
LD analysis of UCP1 gene variants identified in the 3′UTR and 5′UTR regions based on data from the LDpair tool. LD blocks are marked by colors.

**Table 1 genes-14-00789-t001:** Patient and control characteristics.

	Controls (*n* = 36)	MetS Patients		
		All (*n* = 59)	with T2DM (*n* = 29)	w/o T2DM (*n* = 30)
**Sex** (females/males)	18/18	32/27	17/12	15/15
Age	36.33 ± 13.03	61.5 ± 12.91	63 ± 14.06	60 ± 11.49
Total cholesterol [mg/dL]	-	192.82 ± 51.52	186.94 ± 55.45	198.69 ± 46.34
LDL [mg/dL]	-	102.36 ± 41.88	101.16 ± 40.18	103.69 ± 43.73
HDL [mg/dL]				
- women	-	53.53 ± 17.51	51.72 ± 21.58	55.71 ± 11.17
- men	-	45.81 ± 12.35	43.65 ± 15.75	47.53 ± 9.01
Triglycerides [mg/dL]	-	242.46 ± 188.70	254.25 ± 246.92	231.07 ± 110.24
Fasting blood glucose [mg/dL]	-	128.13 ± 51.01	151.80 ± 57.17	106.03 ± 32.01
Body fat percentage [%]	23.89 ± 6.12	-	-	-
BMI	22.77 ± 1.86	34.29 ± 7.55	33.96 ± 9.61	34.60 ± 4.76
Waist circumference [cm]				
- women	75.24 ± 5.82	111.30 ± 19.52	111.82 ± 23.49	110.70 ± 14.59
- men	84.11 ± 5.53	110.29 ± 12.54	105.04 ± 12.81	114.48 ± 10.99
WHR				
- women	0.78 ± 0.03	0.97 ± 0.13	0.97 ± 0.13	0.98 ± 0.12
- men	0.85 ± 0.05	1.07 ± 0.09	1.06 ± 0.05	1.08 ± 0.12
Systolic blood pressure [mmHg]	120.31 ± 7.26	142.73 ± 19.86	134.86 ± 18.90	150.33 ± 17.95
Diastolic blood pressure [mmHg]	76.77 ± 6.07	83.41 ± 12.61	77.34 ± 11.73	89.27 ± 10.62

**Table 2 genes-14-00789-t002:** Selected variants of the *UCP1* gene with the different frequencies of alleles and genotypes between controls and MetS patients.

SNV Position			Control		MetS					
GRCh38/hg38	rsID	Variable	N	%	N	%	OR *	CI95%		*p* Value
140558375	**rs77149926**	C	55	88.71	100	98.04	1			0.011
		T	7	11.29	2	1.96	0.184	0.04	0.80	
		CC	25	80.65	49	96.08	1			0.023
		CT+TT	6	19.35	2	3.92	0.198	0.04	0.92	
140560701	**unknown**	T	68	97.14	75	87.21	1			0.026
		C	2	2.857	11	12.79	4.174	1.02	17.03	
		TT	33	94.29	32	74.42	1			0.020
		CT+CC	2	5.71	11	25.58	4.742	1.11	20.22	
140561713	**rs183105785**	A	69	98.57	77	91.67	1			0.055
		C	1	1.43	7	8.33	4.484	0.75	26.66	
		AA	34	97.14	35	83.33	1			0.049
		AC+CC	1	2.86	7	16.67	4.859	0.79	29.83	
140563477	**unknown**	T	69	98.57	79	91.86	1			0.060
		C	1	1.43	7	8.14	4.371	0.74	25.98	
		TT	34	97.14	36	83.72	1			0.054
		CT+CC	1	2.86	7	16.28	4.726	0.77	28.99	
140569304	**rs36207410**	G	55	88.71	100	98.04	1			0.011
		A	7	11.29	2	1.96	0.184	0.04	0.80	
		GG	25	80.65	49	96.08	1			0.023
		AG+AA	6	19.35	2	3.92	0.198	0.04	0.92	
140570776	**rs36207408**	G	55	88.71	99	97.06	1			0.031
		A	7	11.29	3	2.94	0.260	0.07	0.97	
		GG	25	80.65	48	94.12	1			0.060
		AG+AA	6	19.35	3	5.88	0.283	0.07	1.13	
140571476	**rs76129861**	G	55	88.71	99	97.06	1			0.031
		A	7	11.29	3	2.94	0.260	0.07	0.97	
		GG	25	80.65	48	94.12	1			0.060
		AG+AA	6	19.35	3	5.88	0.283	0.07	1.13	
140572036	**rs56724370**	A	55	88.71	99	97.06	1			0.031
		C	7	11.29	3	2.94	0.260	0.07	0.97	
		AA	25	80.65	48	94.12	1			0.060
		AC+CC	6	19.35	3	5.88	0.283	0.07	1.13	
140572113	**rs77178927**	T	55	88.71	99	97.06	1			0.031
		G	7	11.29	3	2.94	0.260	0.07	0.97	
		TT	25	80.65	48	94.12	1			0.060
		GT+GG	6	19.35	3	5.88	0.283	0.07	1.13	

* Due to the difference in the age factor between controls and patients, the odd ratio here may not reflect true effects of these genetic variations.

**Table 3 genes-14-00789-t003:** Selected variants of the UCP1 gene with the different frequencies of alleles and genotypes between controls and MetS with T2DM patients.

SNV Position			Control		T2DM					
GRCh38/hg38	rsID	Variation	N	%	N	%	OR *	CI95%		*p* Value
140558375	**rs77149926**	C	55	88.71	53	98.15	1			0.046
		T	7	11.29	1	1.85	0.207	0.03	1.25	
		CC	25	80.65	26	96.30	1			0.070
		CT+TT	6	19.35	1	3.70	0.222	0.03	1.42	
140560701	**unknown**	T	68	97.14	21	75.00	1			0.001
		C	2	2.86	7	25.00	9.558	2.11	43.32	
		TT	33	94.29	7	50.00	1			0.001
		CT+CC	2	5.71	7	50.00	13.4	2.61	68.79	
140561203	**unknown**	G	64	91.43	20	71.43	1			0.011
		A	6	8.57	8	28.57	4.114	1.32	12.81	
		GG	29	82.86	6	42.86	1			0.006
		AG+AA	6	17.14	8	57.14	5.935	1.57	22.40	
140561340	**unknown**	T	58	82.86	16	57.14	1			0.008
		A	12	17.14	12	42.86	3.545	1.36	9.22	
		TT	23	65.71	2	14.29	1			0.001
		AT+AA	12	34.29	12	85.71	9.400	2.05	43.04	
140561713	**rs183105785**	A	69	98.57	25	89.29	1			0.037
		C	1	1.43	3	10.71	6.359	0.89	45.41	
		AA	34	97.14	11	78.57	1			0.034
		AC+CC	1	2.86	3	21.43	7.000	0.92	53.08	
140563477	**unknown**	T	69	98.57	23	82.14	1			0.002
		C	1	1.43	5	17.86	10.84	1.68	70.01	
		TT	34	97.14	9	64.29	1			0.002
		CT+CC	1	2.86	5	35.71	13.32	1.91	92.94	
140564690	**unknown**	T	55	78.57	15	53.57	1			0.014
		C	15	21.43	13	46.43	3.119	1.24	7.84	
		TT	20	57.14	1	7.14	1			0.002
		CT+CC	15	42.86	13	92.86	11.90	1.95	72.83	
140567294	**unknown**	A	63	90.00	21	75.00	1			0.056
		C	7	10.00	7	25.00	2.953	0.96	9.09	
		AA	28	80.00	7	50.00	1			0.038
		AC+CC	7	20.00	7	50.00	3.800	1.04	13.84	
140568108	**rs3138733**	CGTGTGTGT	44	62.86	23	82.14	1			0.065
	**GTGTGTGT**	C	26	37.14	5	17.86	0.393	0.14	1.12	
	**insertion**	(CGTGTGTGT)_2_	14	40.00	10	71.43	1			0.049
		CCGTGTGTGT+CC	21	60.00	4	28.57	0.289	0.08	1.05	
140568337	**unknown**	A	52	74.29	15	53.57	1			0.047
		T	18	25.71	13	46.43	2.472	1.00	6.09	
		AA	19	54.29	1	7.143	1			0.003
		AT+TT	16	45.71	13	92.86	10.64	1.74	64.98	
140569265	**rs3811787**	T	57	91.94	42	77.78	1			0.032
		G	5	8.07	12	22.22	3.075	1.05	9.04	
		TT	26	83.87	17	62.96	1			0.072
		GT+GG	5	16.13	10	37.04	2.891	0.87	9.55	
140569304	**rs36207410**	G	55	88.71	53	98.15	1			0.046
		A	7	11.29	1	1.85	0.207	0.03	1.25	
		GG	25	80.65	26	96.3	1			0.070
		AG+AA	6	19.35	1	3.70	0.222	0.03	1.42	
140573371	**rs6536992**	G	55	88.71	39	72.22	1			0.024
		A	7	11.29	15	27.78	2.904	1.11	7.60	
		GG	25	80.65	16	59.26	1			0.077
		AG+AA	6	19.35	11	40.74	2.734	0.87	8.58	
140573392	**rs6536993**	C	55	88.71	39	72.22	1			0.024
		T	7	11.29	15	27.78	2.904	1.11	7.60	
		CC	25	80.65	16	59.26	1			0.077
		CT+TT	6	19.35	11	40.74	2.734	0.87	8.58	
140573493	**rs6536994**	A	55	88.71	39	72.22	1			0.024
		T	7	11.29	15	27.78	2.904	1.11	7.60	
		AA	25	80.65	16	59.26	1			0.077
		AT+TT	6	19.35	11	40.74	2.734	0.87	8.58	

* Due to the difference in the age factor between controls and patients, the odd ratio here may not reflect true effects of these genetic variations.

**Table 4 genes-14-00789-t004:** Characteristics of selected *UCP1* gene variants and variations differently distributed in the whole group of MetS patients and MetS with T2DM in comparison to controls.

rsID	Position (GRCh38.p13)	Location	Allele Frequency	Studied Previously?	Conservation (Score)	MetS **	MetS ** +T2DM
**rs77149926**	chr4:140558375	3′UTR	C = 93.93%	No	−0.627	+	+
unknown	chr4:140560701	Intron 5	-	No	-	+	+
unknown	chr4:140561203	Intron 5	-	No	-	-	+
unknown	chr4:140561340	Intron 5	-	No	-	-	+
**rs183105785**	chr4:140561713	Intron 5	A ~ 100%	No	0.328	+	+
unknown	chr4:140563477	Exon 3	-	No	-	+	+
unknown	chr4:140564690	Intron 2	-	No	-	-	+
unknown	chr4:140567294	Intron 2	-	No	-	-	+
**rs3138733** (Indel)	chr4:140568109-140568156	Intron 1	(GT)_24_ = 12.14%	No	2.089	-	+
unknown	chr4:140568337	Intron 1	-	No	-	-	+
**rs3811787**	chr4:140569265	5′UTR	T = 74.97%	Yes *	−2.682	-	+
**rs36207410**	chr4:140569304	5′UTR	G = 93.38%	No	−2.179	+	+
**rs36207408**	chr4:140570776	5′UTR	G = 93.72%	No	−0.223	+	-
**rs76129861**	chr4:140571476	5′UTR	G = 93.70%	No	0.005	+	-
**rs56724370**	chr4:140572036	5′UTR	A = 93.70%	No	−0.244	+	-
**rs77178927**	chr4:140572113	5′UTR	T = 98.18%	No	0.042	+	-
**rs6536992**	chr4:140573371	5′UTR	G = 68.23%	No	−0.639	-	+
**rs6536993**	chr4:140573392	5′UTR	C = 67.31%	No	−0.372	-	+
**rs6536994**	chr4:140573493	5′UTR	A = 78.14%	No	−1.211	-	+

* Cha et al. 2008 [30], Jin et al. 2020 [31], Nikanorova et al. 2021 [32]; ** potentially associated with susceptibility to disorder, + yes; - not associated.

## Data Availability

The data obtained in this study are presented within the article/Appendix A. Additional data are available upon reasonable request from the corresponding author.

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
