# Peer review of "NGS Sequencing Reveals New UCP1 Gene Variants Potentially Associated with MetS and/or T2DM Risk in the Polish Population—A Preliminary Study"

_genes, 2023, doi:10.3390/genes14040789_

Round 1
Reviewer 1 Report
This manuscript (genes-2139488) aims to study what UCP1 gene variants may associate with metabolic syndrome (MetS) or this disease with type II diabetes mellitus (MetS with T2DM) in Polish population. There were 59 MetS patients, including twenty-nine with MetS with T2DM, and thirty-six control individuals recruited to this study. Targeted NGS sequencing of UCP1 gene showed nine variants associated with MetS risk and fifteen variants with the altered risk in MetS with T2DM. There are four tables and two figures, with which the authors suggested that all these variations except one are new in the UCP1 research. Overall, there are several key issues that need to be addressed.
First, the introduction does not have sufficient background and all relevant references. Particularly, those should be included regarding the prevalence and the clinical features of MetS in Polish population, such as Raposo (2021) and Rajca et al. (2021) in the journal Pol Arch Intern Med. These results can be particularly important in the understanding of the current results. Also, some paragraphs not relevant to the subject are redundant.
Second, the results and discussion can be more organized. It is not conventional to have the first several paragraphs dedicated to the discussion that are not relevant to the current results. Also, the sentences that summarize the current findings can be used as the headlines of the sections.
Third, the authors need to address whether the odd ratios were in the expected range when this control group was used. Age is a key factor in the MetS progression, and MetS has a high prevalence (>30%) in the entire population. While young adults indeed have a lower prevalence, they still have a high chance to develop MetS. Based on the odd ratios in this study, this control group had the chance of ~20% to have MetS later. The authors may address whether such numbers are consistent with the study design based on the information in the references addressed above.
Fourth, it is paradoxical to see that not all the nine variants identified in MetS studies were included in the list of variants associated with MetS with T2DM. For example, rs36207408, rs76129861, rs56724370 and rs77178927 were not in the 15-variant list. Paragraphs addressing this discrepancy should be provided.
Reviewer 2 Report
1. Include recent statistics
2. Non readable sentences
3. Grammatical mistakes
Additional Comments:
1. What is the main question addressed by the research?
Ans: The authors had performed targeted panel NGS sequencing of the whole UCP1 gene in 59 metabolic syndrome (MetS patients) including 29 patients with T2DM, and 36 controls using the MiSeq platform seeking new promising research targets as MetS and/or T2DM prediction markers. They have identified 12 variants among which only rs3811787 has been investigated previously by others.2. Do you consider the topic original or relevant in the field? Does it address a specific gap in the field? Yes. It is original. 3. What does it add to the subject area compared with other published material? Ans: Though UCP1 is already reported to be associated with different populations, this paper reports it in the Polish population. Can be used as a therapeutic target. 4. What specific improvements should the authors consider regarding the methodology? What further controls should be considered? Ans: What I have understood from the Grant number is that the study is an old one or sample collections are not done very recently. If I understand it correctly, the addition of new control may not be practically feasible. 5. Are the conclusions consistent with the evidence and arguments presented and do they address the main question posed? Ans: Yes 6. Are the references appropriate? Ans: Reference need to update 7. Please include any additional comments on the tables and figures. NIL
Round 2
Reviewer 1 Report
Please see the attached document.

Author Response
Dear Reviewer,
please find attached replay for all your comments

Round 3
Reviewer 1 Report
This revision (genes-2139488) has incorporated the necessary changes, mostly in accordance with the given suggestions. The current version of the manuscript is of good quality for publication, but there are still some suggestions that could enhance it further.
Regarding Tables 2 and 3, the authors may consider adding an asterisk to each OR in the front row. Also, it may be helpful to include a sentence in the title of the two tables, such as "Due to the difference in the age factor between controls and patients, the odd ratio here may not reflect true effects of these genetic variations."
In Line 267, Page 11, the authors may want to change "3.2. LD analyses" to "3.2. Genetic variants in 5’-UTR could be grouped into two LD blocks with suppressive or enhancing effects on MetS and MetS with T2DM, respectively."
On Line 347, Page 14, "thymine" should be changed to "threonine." Based on the predicted structure, Thr-123 is likely in one of the transmembrane α helices. The change into Pro, a known α helix breaker, may have some effects on membrane interaction.
Regarding Line 389, Page 14, the two sentences "The third limitation … group" can be revised as follows: "The third limitation is the mismatch of ages between the older MetS and the younger control groups. Fortunately, the potential of the control subjects to develop MetS later in their lives should lead to underestimation of their enhancing or protective effects, which therefore should leave our conclusions largely intact."
Author Response
The response to the comments provided by the Reviewer 3
This revision (genes-2139488) has incorporated the necessary changes, mostly in accordance with the given suggestions. The current version of the manuscript is of good quality for publication, but there are still some suggestions that could enhance it further.
We thank the Reviewer for finding our manuscript of good quality. We tried to do our best to improve it according to the Reviewer suggestions.
Regarding Tables 2 and 3, the authors may consider adding an asterisk to each OR in the front row. Also, it may be helpful to include a sentence in the title of the two tables, such as "Due to the difference in the age factor between controls and patients, the odd ratio here may not reflect true effects of these genetic variations."
According to suggestion we added * in OR column and the following sentence below tables 2 and 3 "Due to the difference in the age factor between controls and patients, the odd ratio here may not reflect true effects of these genetic variations."
In Line 267, Page 11, the authors may want to change "3.2. LD analyses" to "3.2. Genetic variants in 5’-UTR could be grouped into two LD blocks with suppressive or enhancing effects on MetS and MetS with T2DM, respectively."
We appreciate the Reviewer suggestion, however this subtitle regards not only to 5’ analysis but for entire gene.
What is more provided analysis was not performed on our real data but on the basis of the LDpair tool, only for the identified UCP1 gene variants (established rsIDs) with a minor allele frequency (MAF) higher than 1% in the European population.
However, we performed LD analysis with our genotyping data for controls for SNVs associated with MetS and/ or MetS with T2DM using SHEsis software (http://analysis.bio-x.cn/myAnalysis.php available access day 13.03.20230) and we have added additional data to this Subpart of the Results
On Line 347, Page 14, "thymine" should be changed to "threonine." Based on the predicted structure, Thr-123 is likely in one of the transmembrane α helices. The change into Pro, a known α helix breaker, may have some effects on membrane interaction.
We have to apologize since I our analysis we made a mistake in the discussion. As is correctly in tables for this variation (chr4:140563477) there is the exchange from T to C, not as we mentioned in the discussion T to G. Therefore, the correct amino acid should be alanine. Thanks to the Reviewer suggestion we checked once again our results and it allowed us to correct the mistake.
Regarding Line 389, Page 14, the two sentences "The third limitation … group" can be revised as follows: "The third limitation is the mismatch of ages between the older MetS and the younger control groups. Fortunately, the potential of the control subjects to develop MetS later in their lives should lead to underestimation of their enhancing or protective effects, which therefore should leave our conclusions largely intact."
According to the Reviewer suggestion we changed the paragraph about the third limitation.
